# Cytokine Responses during *Mycobacterium tuberculosis* H37Rv and *Ascaris lumbricoides* Costimulation Using Human THP-1 and Jurkat Cells, and a Pilot Human Tuberculosis and Helminth Coinfection Study

**DOI:** 10.3390/microorganisms11071846

**Published:** 2023-07-21

**Authors:** Khethiwe N. Bhengu, Ravesh Singh, Pragalathan Naidoo, Miranda N. Mpaka-Mbatha, Nomzamo Nembe-Mafa, Zilungile L. Mkhize-Kwitshana

**Affiliations:** 1Department of Medical Microbiology, School of Laboratory Medicine and Medical Sciences, College of Health Sciences, Nelson R. Mandela School of Medicine, University of KwaZulu-Natal, Durban 4001, South Africa; bhengukn@mut.ac.za (K.N.B.);; 2Division of Research Capacity Development, South African Medical Research Council (SAMRC), Cape Town 7505, South Africa; 3Department of Biomedical Sciences, Faculty of Natural Sciences, Mangosuthu University of Technology, Umlazi, Durban 4031, South Africa; 4Department of Medical Microbiology, School of Laboratory Medicine and Medical Sciences, College of Health Sciences, Howard College, University of KwaZulu-Natal, Durban 4041, South Africa

**Keywords:** *Mycobacterium tuberculosis* H37Rv, *Ascaris lumbricoides* excretory-secretory proteins, Jurkat cells, THP-1 cells, human tuberculosis and helminth co-infection, cytokine gene transcription levels

## Abstract

Background: Helminth infections are widespread in tuberculosis-endemic areas and are associated with an increased risk of active tuberculosis. In contrast to the pro-inflammatory Th1 responses elicited by *Mycobacterium tuberculosis* (Mtb) infection, helminth infections induce anti-inflammatory Th2/Treg responses. A robust Th2 response has been linked to reduced tuberculosis protection. Several studies show the effect of helminth infection on BCG vaccination and TB, but the mechanisms remain unclear. Aim: To determine the cytokine response profiles during tuberculosis and intestinal helminth coinfection. Methods: For the in vitro study, lymphocytic Jurkat and monocytic THP-1 cell lines were stimulated with Mtb H37Rv and *Ascaris lumbricoides* (*A. lumbricoides*) excretory-secretory protein extracts for 24 and 48 h. The pilot human ex vivo study consisted of participants infected with Mtb, helminths, or coinfected with both Mtb and helminths. Thereafter, the gene transcription levels of IFN-γ, TNF-α, granzyme B, perforin, IL-2, IL-17, NFATC2, Eomesodermin, IL-4, IL-5, IL-10, TGF-β and FoxP3 in the unstimulated/uninfected controls, singly stimulated/infected and costimulated/coinfected groups were determined using RT-qPCR. Results: TB-stimulated Jurkat cells had significantly higher levels of IFN-γ, TNF-α, granzyme B, and perforin compared to unstimulated controls, LPS- and *A. lumbricoides*-stimulated cells, and *A. lumbricoides* plus TB-costimulated cells (*p* < 0.0001). IL-2, IL-17, Eomes, and NFATC2 levels were also higher in TB-stimulated Jurkat cells (*p* < 0.0001). Jurkat and THP-1 cells singly stimulated with TB had lower IL-5 and IL-4 levels compared to those singly stimulated with *A. lumbricoides* and those costimulated with TB plus *A. lumbricoides* (*p* < 0.0001). *A. lumbricoides*-singly stimulated cells had higher IL-4 levels compared to TB plus *A. lumbricoides*-costimulated Jurkat and THP-1 cells (*p* < 0.0001). TGF-β levels were also lower in TB-singly stimulated cells compared to TB plus *A. lumbricoides*-costimulated cells (*p* < 0.0001). IL-10 levels were lower in TB-stimulated Jurkat and THP-1 cells compared to TB plus *A. lumbricoides*-costimulated cells (*p* < 0.0001). Similar results were noted for the human ex vivo study, albeit with a smaller sample size. Conclusions: Data suggest that helminths induce a predominant Th2/Treg response which may downregulate critical Th1 responses that are crucial for tuberculosis protection.

## 1. Introduction

Tuberculosis (TB) infection is caused by *Mycobacterium tuberculosis* (Mtb), a significant global health challenge and one of the deadliest diseases caused by a single infectious agent [1]. Ten million TB cases and 1.4 million fatalities were reported globally in 2020 [1]. Furthermore, a quarter of the global population is latently infected with TB [1]. A competent immune system contains the TB infection in an asymptomatic/latent state. However, there are underlying factors in 5–10% of hosts that may lead to the development of active TB from latent infection [1,2]. 

Helminths infect 1.5 billion people worldwide, and *Ascaris lumbricoides* (*A. lumbricoides*), the most prevalent helminth, infects an estimated 807 million–1.2 billion people worldwide [3]. Humans are infected through ingestion of embryonated *A. lumbricoides* eggs containing larvae [3]. The hatched larvae enter the circulation and migrate to the lungs causing pneumonitis and eosinophilia [3]. Larvae mature further in the lungs (10 to 14 days), penetrate the alveolar walls, ascend the bronchial tree to the throat where they are coughed up and swallowed, thereby re-entering the gastrointestinal tract where they mature in the small intestines [3]. There is a significant geographic overlap between TB and helminth infection, particularly in low and middle-income countries (LMICs), with 20–35% of people being co-infected [4]. The impact of helminths on cell-mediated immunity has been the subject of numerous investigations [5,6,7,8,9]. However, it is still unclear if parasite infection is associated with TB activation from a dormant condition to active disease development [5]. 

An efficient T-helper type 1 (Th1)/pro-inflammatory response is required to control intracellular Mtb [7,10]. The Th1/pro-inflammatory response is characterised by the production of interferon-gamma (IFN-γ), tumour necrosis factor-alpha (TNF-α), and interleukins (IL-1, IL-6 and IL-12) [11]. In contrast, helminths skew the immunity towards a predominant T-helper type 2 (Th2)/anti-inflammatory and Regulatory (Treg) response, leading to the release of IL-4, IL-5, IL-9, IL-10, IL-13, and transforming growth factor-beta (TGF-β) [10,12]. These two arms of immune responses counter-regulate each other. Subsequently, helminths have been shown to reduce Bacille Calmette–Guerin (BCG) immunogenicity [13,14], weaken Mtb-specific Th1 responses, downregulate co-stimulatory molecules [15], induce anergy [16], and reduce treatment response, particularly in pulmonary TB [17,18].

However, in some studies, helminths were demonstrated to have no impact on human tuberculin skin tests [19], Mtb infection [20], or the improvement of TB disease management [20]. Therefore, reports on TB immune responses in cases of helminth coinfection are variable and dependent on the infecting parasite and the type of study [8,9,21]. Studies involving *Nippostrongylus brasiliensis* (Nb) and mycobacterial coinfection in mice yielded divergent findings on Mtb growth control. One study determined that mycobacterial clearance in the lungs of tuberculosis and Nb-coinfected mice was not delayed and that the helminth-induced Th2 responses do not exacerbate tuberculosis infection [22]. It was also reported that early-stage Nb infection increased macrophage production, which confers protection against subsequent stages of mycobacterial disease [23]. Conversely, another study reported that mycobacterial burden was higher in tuberculosis and Nb-coinfected mice and that these animals had reduced resistance to TB infection [24]. In human studies, *A. lumbricoides* infection was associated with negative tuberculin skin tests in children, suggestive of poor tuberculosis immune response [25,26]. 

Therefore, the effect of different helminth species and their antigens on immunity, particularly on macrophages, the primary effector cells in tuberculosis infection, remains unclear. Hence, the present study compared the cytokine immune responses in human THP-1 and Jurkat cells stimulated with and without coincident tuberculosis and *A. lumbricoides* antigen to simulate coinfection. The study was also extended to humans to determine the cytokine immune responses in ex vivo data. The detailed abbreviations and definitions used in the paper are listed in Table 1.

## 2. Materials and Methods

### 2.1. Part 1: In Vitro Studies

#### 2.1.1. Bacterial Cultures

The H37Rv strain of Mtb (bacterial strain number 25618) was purchased from the American Type Culture Collection (ATCC) through Thistle QA Laboratory Services Cc in Johannesburg, South Africa (SA). H37Rv was cultured to log phase at 37 °C in 5% CO_2_ in Middlebrook 7H9 broth with 0.05% Tween-80 and 10% oleic acid albumin dextrose catalase enrichment (OADC) (Becton Dickinson). Colony-forming units were counted by serial dilutions on Middlebrook agar plates. The protein concentration of the H37Rv was determined using the Bradford assay [27] and an optimal concentration of 5 µg/mL was used for cell stimulation. Cells were preserved in 1 mL aliquots at −80 °C until further use.

#### 2.1.2. Helminth (*A. lumbricoides*) Excretory-Secretory Protein Extracts 

Whole worm excretory-secretory protein (ESP) extracts of *A. lumbricoides*, kindly donated by Prof William Horsnell, were prepared and supplied by the Division of Immunology, Department of Pathology from the Faculty of Health Sciences at the University of Cape Town, SA. Adult worms were obtained from patients from the Red Cross War Memorial Children’s Hospital (Cape Town, South Africa), and were used to acquire *A. lumbricoides* excretory proteins. The *A. lumbricoides* excretory proteins were obtained by keeping the worms alive at 37 °C in Dulbecco modified essential medium with 1% Pen-strep (Thermofisher Scientific, Waltman, MA, USA), and 1% glucose (*wt.*/*vol*). The media was collected three times a day. Using Amicon ultra concentrator, extract proteins were concentrated and resuspended in 5 mL of phosphate-buffered saline (Merck). All antigens were measured for protein content with a BCA protein estimation kit (Thermofisher Scientific) or by using the Bradford assay previously described [27] and stored at −80 °C at a standard concentration of 500 µg/mL until further use. 

#### 2.1.3. Cell Culture and Treatment

Human monocytic THP-1 (lot number: TIB-202) and lymphocytic Jurkat (lot number TIB-152) cells were purchased from the ATCC by Thistle QA Laboratory Services Cc in Johannesburg, SA. The cells were maintained in 25 cm^3^ cell culture flasks containing Roswell Park Memorial Institute (RPMI) supplemented with 2 mM L-glutamine, 5% HEPES, 100 U/mL penicillin, 100 µg/mL streptomycin, and 10% foetal bovine serum (FBS) at 37 °C in a humidified atmosphere of 5% CO_2._

Thereafter, the Jurkat and THP-1 cells were aliquoted into the 24 well multi-well plates in 1 mL aliquots (>1 × 10^6^) and unstimulated or stimulated with either lipopolysaccharide (LPS) (Thermofisher—catalogue number 00-4976-93), Mtb H37Rv, or *A. lumbricoides* ESP extracts. The unstimulated cells served as the negative control group, the LPS-stimulated group received 1 mg/mL LPS and served as a positive control, the *A. lumbricoides*-singly stimulated group were stimulated with 5 µ/mL of *A. lumbricoides* excretory protein extracts only, the Mtb-singly stimulated group were stimulated with 5 µg/mL of Mtb H37Rv only, and lastly, the costimulated group were co-stimulated with both 5 µg/mL of *A. lumbricoides* excretory protein ESP extracts and 5 µg/mL of Mtb H37Rv. Two independent experiments were set up in triplicate. Thereafter, the unstimulated/stimulated Jurkat and THP-1 cells were incubated for 24 or 48 h at 37 °C. At the end of the incubation period, the cells were collected, stored in Trizol^®^ (Invitrogen; Thermo Fisher Scientific, Inc. catalogue 15596026) and stored in the −80 °C freezer for RNA extraction and gene transcription levels studies using Quantitative PCR.

#### 2.1.4. Real-Time-Quantitative PCR (RT-qPCR)

RNA was extracted from unstimulated/stimulated Jurkat and THP-1 cell lines using the Trizol^®^ reagent (Invitrogen; Thermo Fisher Scientific, Inc., Waltham, MA, USA, catalogue 15596026) and the Pure Link^TM^ RNA Mini Kit (Thermofisher Scientific, catalogue number 12183018A). The total RNA had to be DNA-free, therefore Pure link^®^ DNase treatment at 80 µL per sample was used. The DNase treatment included 88 µL DNase buffer, 110 µL resuspended DNase, and 620 µL RNase-free water. The prepared DNase mixture was added directly onto the surface of the spin cartridge membrane, incubated at 15 min, washed with buffer, ethanol was added, and the cartridge was spun. RNase-free water was added to the spin cartridge and incubated for 1 min. The spin cartridge was spun with the recovery tube. The RNA preparation was added to a Nanodrop 2000 spectrophotometer (Thermofisher Scientific) to check for purity and concentration. Thereafter, the isolated RNA was reverse transcribed to cDNA using the High-Capacity cDNA Reverse Transcription Kit with RNase Inhibitor (Thermofisher Scientific, catalogue number 4374966), as per the manufacturer’s instructions and reaction protocol. The Nanodrop 2000 spectrophotometer (Thermofisher Scientific) was used to quantify the total cDNA. The cDNA samples with an optical density at 260/280 nm (OD_260/280_) >1.8 were used for RT-qPCR. 

The Applied Biosystems Quant Studio 5 PCR instrument and software (Thermofisher Scientific, Waltham, MA, USA) were used to determine the transcription levels of the cytokine genes of interest listed in Table 2 in the unstimulated (control cells), tuberculosis-stimulated, *A. lumbricoides*-stimulated, LPS-stimulated, and *A. lumbricoides* and tuberculosis-co-stimulated cells. 

The PCR master mix was prepared by adding 5 µL PCR-grade water (Thermofisher Scientific, catalogue number 10977023), 0.50 µL FAM-labelled cytokine probe mix (Thermofisher Scientific) (Table 2), 2.50 µL Fast Start 4× probe master mix (Thermofisher, catalogue number A15300) and 2 µL cDNA to make a total of 10 µL per sample. Glyceraldehyde 3-diphosphate dehydrogenase (GAPDH) was used as a housekeeping gene. PCR-grade water (Thermofisher Scientific, catalogue number 10977023), instead of cDNA, was used as a negative control. 

The PCR was performed at 95 °C for 1 min, followed by 45 cycles comprising denaturation at 95 °C for 30 s, annealing at 60 °C for 30 s and extension at 72 °C for 30 s. All PCR reactions were run in duplicate. Data were collected using the Applied Biosystems Quant Studio 5 V.2.3 software (Thermofisher Scientific, Waltham, MA, USA). 

Serial dilutions of pooled cDNA synthesised from the total RNA were performed for each target gene and GAPDH, which served as standard curves for quantitative analysis, ranging from 1 ng/µL to 1000 ng/µL. Gene transcription levels results were depicted as the transcription levels of the gene of interest divided by the transcription levels of GAPDH. 

### 2.2. Part B: Human Ex-Vivo Experiment

The Th-1, Th-17 and Treg cytokine gene and transcription factor’s transcription levels study were also piloted for human ex-vivo experiments to compare the human and in vitro cytokine profile results. The current analysis is a sub-study of a previously described cohort of 414 individuals recruited from 6 primary healthcare clinics in a peri-urban, poor settlement in the eThekwini district of KwaZulu-Natal [28]. In this study, cytokine analysis was undertaken for 164 participants; of those, 96 were HIV-infected and had to be excluded, leaving 68 eligible participants. Thereafter, the eligible individuals were subdivided into uninfected controls (no helminth or TB) (*n* = 18), helminth-singly infected only (*n* = 35), TB-singly infected only (*n* = 6), and TB and helminth-co-infected (*n* = 6) groups. 

Stool samples were collected for microscopical detection of helminth eggs/larvae using the Kato–Katz and Mini Parasep methods. Blood samples were also collected for parasite serology (*A. lumbricoides*-specific IgE and IgG4) to improve the sensitivity and specificity of parasite detection [29]. TB diagnosis and confirmatory results were obtained from the district hospital laboratory that services the clinics where participants were recruited. The sputum was analysed using the GeneXpert Infinity 48 s (Catalog number: Infinity-48).

Whole blood samples (4 mls) collected from the recruited participants were also stored in Trizol^®^ reagent (Invitrogen; Thermo Fisher Scientific, Inc.) at −80 °C for RNA extraction and RT-qPCR-based gene transcription levels studies as described for the in vitro experiments above. 

### 2.3. Statistical Analysis

A standard curve method was used to calculate gene transcription levels, whereby the transcription levels of the target gene were divided by the transcription levels value of the housekeeping gene (GAPDH). Values were expressed as medians. All cytokine gene transcription levels data were analysed using GraphPad Prism 5 (GraphPad Software, Inc., San Diego, CA, USA) statistical software package. For the in vitro and human ex vivo studies, analysis of variance (ANOVA) or the Kruskal–Wallis test with Tukey or Dunn’s Multiple Comparison was used to assess for statistical significance in cytokine gene transcription levels profiles between the different groups (uninfected/unstimulated controls, singly infected/stimulated and coinfected/costimulated groups). Thereafter, the Mann–Whitney or Student’s *t*-test was used to calculate the *p*-value between the two groups. All data presented in figures below are expressed as the median and interquartile range. A *p* < 0.05 was considered statistically significant.

## 3. Results

### 3.1. Part 1: In Vitro Study

Profiling of cytokine and transcription factor gene transcription levels was performed using THP-1 and Jurkat cells to investigate whether TB stimulation would upregulate pro-inflammatory and Th1 cytokines and whether *A. lumbricoides* coinfection would downregulate these. Furthermore, it was aimed to determine whether *A. lumbricoides* would upregulate Th2 and regulatory cytokines. 

#### 3.1.1. Th1/Pro-Inflammatory Immune Responses

Cytokine gene transcription levels levels in unstimulated and stimulated human cell lines are summarised in the figures below, showing a significant increase of Th1/pro-inflammatory cytokine genes after TB stimulation.

IFN-γ and TNF-α (at both 24 and 48 h stimulation time points), granzyme B (24 h stimulation only) and perforin (48 h stimulation only) levels were significantly higher in the TB-singly stimulated Jurkat cells compared to the unstimulated controls, LPS- and *A. lumbricoides*-singly stimulated Jurkat cells, and *A. lumbricoides* plus TB-costimulated Jurkat cells (*p* < 0.0001) (Figure 1). Similar results were noted for the THP-1 stimulated cells, apart from perforin, where similar findings were noted at 24 h and 48 h (Figure 2).

IL-2, IL-17, Eomes, and NFATC2 (at both 24 and 48 h stimulation) were significantly higher in TB-singly stimulated Jurkat cells compared to the unstimulated controls, LPS- and *A. lumbricoides*-singly stimulated Jurkat cells, and TB plus *A. lumbricoides*-costimulated Jurkat cells (*p* < 0.0001) (Figure 3). Similar findings resulted from tests on THP-1 cells (*p* < 0.0001) (Figure 4).

#### 3.1.2. Th2/Anti-Inflammatory, Immune Responses

Type 2 cytokine responses after stimulation of cell lines in Figure 5 show that both IL-4 and IL-5 were increased after *A. lumbricoides*-antigen stimulation of both cell lines.

IL-5 (24 and 48 h stimulation) levels were significantly lower in the TB-singly stimulated cells compared to the *A. lumbricoides*-singly stimulated and TB plus *A. lumbricoides*-costimulated Jurkat and THP-1 cells, which had similar transcription levels (*p* < 0.0001). Similar findings were noted for IL-4; however, the *A. lumbricoides*-singly stimulated cells had significantly higher IL-4 levels compared to the TB plus *A. lumbricoides*-costimulated Jurkat (48 h stimulation) and THP-1 cells (24 and 48 h stimulation) (*p* < 0.0001).

#### 3.1.3. Regulatory Cytokines

Figure 6 illustrates regulatory cytokine transcription levels in the cell lines. In both Jurkat and THP-1cells, both *A. lumbricoides* and *A. lumbricoides* plus TB stimulation significantly increased TGFβ and IL-10 transcription levels at 24 h and remained high at 48 h in *A. lumbricoides*-stimulated cells. FoxP3 was increased in both *A. lumbricoides* and TB-*A. lumbricoides* stimulation in both cell lines and at both time points (24 and 48 h). 

TGF-β levels were significantly lower in the TB-singly stimulated Jurkat cells (24 h stimulation) compared to the TB plus *A. lumbricoides*-costimulated cells, however, the opposite trend was observed for THP-1 cells (24 h stimulation) (*p* < 0.0001). Conversely, no significant differences were noted in Jurkat and THP-1 cells (48 h stimulation) between the TB-singly stimulated and TB plus *A. lumbricoides*-costimulated cells. IL-10 levels were significantly lower in the TB-stimulated Jurkat (24 and 48 h stimulation) and THP-1 (24 h stimulation) cells compared to the TB plus *A. lumbricoides*-costimulated cells (*p* < 0.0001). FoxP3 levels were also significantly lower in the TB-singly stimulated Jurkat and THP-1 cells (24 and 48 h stimulation) in comparison to the TB plus *A. lumbricoides*-costimulated cells (*p* < 0.0001). (Figure 6).

### 3.2. Part 2: Human Ex-Vivo Experiment Results

A total of 414 participants were recruited in the main study [29]; of those, a subpopulation of 164 were eligible for cytokine gene transcription level analysis, based on blood sample availability. However, 96 were HIV-infected and were excluded, leaving 68 eligible participants. Of the eligible participants, 18 were uninfected and were used as controls; 35 were helminth-infected (24 were infected with *A. lumbricoides*, 3 *Trichuris trichiura*, 3 *Taenia* spp., 3 *Schistosoma* spp., and 2 had *Stronglyloides* spp.), 6 had TB, and another 6 had TB and helminth (3 had *A. lumbricoides*, 1 *Schistosoma* spp., 1 *Trichuris trichiura* and 1 with *Taenia* spp.) coinfection.

Regardless of the small sample sizes for these two groups, the Th1/pro-inflammatory cytokines (IFN-γ, TNF-α, granzyme B, IL-2, and IL-17), critical cytokines for TB, were significantly higher among the TB-singly infected individuals compared to the uninfected controls and helminth-infected groups (Figure 5). In the presence of helminth and TB coinfection, these cytokines were decreased, although there was no significant difference noted betweenthe coinfected group and the TB-singly infected group, except for granzyme B, where the TB and helminth-coinfected group had lower levels compared to the TB-singly infected group (Figure 7).

Eomes and NFATC2 were significantly higher in the control group compared to the coinfected group. The coinfected group also had lower Eomes and NFATC2 levels compared to the TB-singly infected and helminth-singly infected groups (Figure 8). IL-4 and IL-10 responses were variably increased in the helminth-infected individuals. TGF-β levels were variably increased in the controls and decreased in TB-singly infected and the helminth and TB-coinfected individuals. FoxP3 levels also differed between the controls and the TB-singly infected groups and between the helminth-singly infected and TB-singly groups. The low number of TB-infected individuals resulted in even lower numbers of the coinfected groups, thus making statistically valid analytical comparisons difficult (Figure 9).

IFN-γ, TNF-α, and IL-17 were highest in TB-infected group (albeit, there were only six individuals). Perforin was similar across all groups, while granzyme B levels differed between the control and coinfected groups (*p* < 0.0001), between the helminth-infected and coinfected groups (*p* = 0.0020), and between TB-infected and coinfected groups (*p* < 0.0001). IL-2 levels differed between the control and the TB plus helminth-coinfection group (*p* < 0.0001) and also between the helminth-infected and coinfected group (*p* = 0.0067).

Eomes levels were higher in the controls than in the TB/helminth co-infected (*p* < 0.0001) and higher in the TB-infected compared to the coinfected individuals (*p* < 0.0001). NFATC2 levels were significantly higher among the controls compared to the coinfected individuals (*p* = 0.0003) and higher in the helminth-infected than in the TB/helminth-coinfected individuals (*p* = 0.0032).

IL-4, IL-10, and TGF-β were higher among uninfected controls and helminth-infected individuals (albeit there was a wide distribution in values) compared to the TB-singly infected and coinfected groups (albeit there was a small sample size). TGF-β was lower in the TB-singly infected group and the coinfected group, compared to the controls (*p* = 0.0012 and *p* < 0.0001, respectively). FoxP3 was significantly lower among the TB-infected compared to both the control (*p* < 0.0001) and helminth-infected groups (*p* = 0.0012). 

## 4. Discussion

The present study aimed to determine the profile of cytokines after stimulation of monocytic and lymphoid cells with *A. lumbricoides* and TB antigens to assess whether *A. lumbricoides* infection would decrease the Th1/pro-inflammatory cytokines essential for TB control and increase the Th2/anti-inflammatory and regulatory cytokines. The human ex vivo data were also used to determine the cytokine responses in helminth and TB infection and in cases of helminth/TB coinfection. The Th1 cytokines were increased in TB-stimulated cells/infected individuals and reduced during coinfection. The Th2 and regulatory cytokines were variably increased in dual infection.

The Th1/pro-inflammatory cytokines, IFN-γ and TNF-α, were upregulated in response to TB compared to the *A. lumbricoides* and coinfection stimulation. This finding suggests that Th1/pro-inflammatory cytokines are upregulated by TB and reduced in helminth coinfection. These cytokines are produced more in pro-inflammatory conditions such as TB [7]. The cytokine IFN-γ is essential for protective defence against intracellular infections. IFN-γ is a key modulator of macrophage activation in *Mycobacterium tuberculosis* (*Mtb*) infection [29,30]. 

TNF-α plays a pivotal role in granuloma formation, which is one of the host’s defence mechanisms against TB [31]. According to some studies, TNF-α levels are frequently high in individuals with active TB infection [32,33]. Our analysis also demonstrated a similar pattern. 

The molecules involved in the cell-mediated killing of intracellular pathogens in the pro-inflammatory response included granzyme B and perforin. Granzyme is a serine protease present in the granules of cytotoxic lymphocytes. Perforin and granzyme work together to kill infected cells or target cells by perforating the cell walls leading to disintegration [34]. Natural Killer (NK) and CD8-positive cells primarily produce granzyme B and perforin [34,35]. They attack malignant or infected cells and cause them to undergo apoptosis [35]. Granzyme B and perforin were both increased at 24 h in the TB-stimulated cells and reduced at 48 h, suggesting that they induce apoptosis in infected cells during the early stages of infection. However, it is crucial to note that more experiments such as tunnel assays or flow cytometry should be performed to validate these results for granzyme B and perforin.

The increased levels of IL-2, IL-17, Eomes, and NFATC2 for both the in vitro and ex vivo analyses are in keeping with the pro-inflammatory response. IL-2 is produced by Th1 cells, and it stimulates T-cell proliferation, among other functions. In turn, Th1 cells produce IL-2, which has been found to stimulate cytotoxic T lymphocytes and Th1 cells during intracellular pathogen invasion [36,37]. Compared to uninfected individuals, patients with active TB have been shown to have high IL-2 cytokine levels suggesting that this cytokine plays a protective role [38]. 

IL-17, an inflammatory cytokine released during the early stages of TB infection, is suggested to increase the synthesis of chemokines that aid in the recruitment of cells essential for granuloma formation [39]. Limiting Mtb growth and immunopathology caused by increased IL-17 production requires a balance between Th1 and Th17 immune responses [40]. Overproduction of IL-17 can increase neutrophil recruitment, which can cause tissue damage [40]. A Th1/Th17 balance is required for anti-mycobacterial immunity and immunological disease prevention [40]. It is notable, then, that the current study determined that, in vitro and ex vivo, the Th1/pro-inflammatory responses are higher in TB infection and reduced in helminth infection and TB/helminth coinfection cases. 

Eomes levels were increased in our study, in both the in vitro and ex vivo experiments in the TB stimulated/infected group. Eomes was increased since it plays a role in the differentiation of cytotoxic T cells [39], which promote the killing of infected cells through the release of granzyme B and perforin [39]. Hence, in our study, Eomes was increased in both the in vitro and ex vivo experiments in the TB-stimulated/infected groups. NFATC2 overexpression aids cell defence against oxidative stress and electrophilic offences by stimulating synthesis of antioxidative and detoxifying enzymes [41]. As expected, pro-inflammatory/Th1 responses were all increased by TB-antigen stimulation and decreased during helminth coinfection in our study. Furthermore, the current study suggests that the *A. lumbricoides* effect of lowering the pro-inflammatory/Th1 cytokine responses to TB could be detrimental to TB control during TB and helminth coinfection.

In the present study, the Th2/anti-inflammatory cytokines, IL-4 and IL-5, were higher in the helminth infection and coinfection in vitro stimulations compared to the TB stimulation. This is in keeping with the Th2-predominant immune response produced by the extracellular helminths. IL-4 was increased at 24 h for both cell lines in the *A. lumbricoides* and coinfection stimulations; however, this was not sustained at 48 h. The upregulation of IL-4 was shown by the significant differences between the *A. lumbricoides* stimulation and the *A. lumbricoides*/TB co-stimulation. High IL-4 downregulates IFN-γ, which may be deleterious for TB control [42]. IL-5- was elevated in the helminth and coinfection group versus the control group. High levels of IL-5 are commonly observed in intestinal helminth- and protozoa-infected hosts, and it also induces eosinophilia, another common manifestation of parasite infection [43]. The current study confirmed the association between *A. lumbricoides* and Th2 cytokine responses. 

The regulatory cytokines, IL10 and transcription factor FoxP3, were upregulated in the helminth-infected and coinfected cells compared to the TB-infected group. The increase is expected in *A. lumbricoides*-treated cells since helminths polarise immunity towards a Th2 and regulatory immune response [14,44]. Transcription factor, FoxP3, was also high for the current study’s in vitro and ex vivo experiments. This upregulation of FoxP3 in helminth- and coinfection-stimulated cells concurs with the study that determined that helminths increase the secretion of TGFβ, which upregulates FoxP3 and promotes differentiation of regulatory cells [45]. Regulatory cytokines, such as IL-10, play a suppressive role in regulating immune homeostasis. Hence, IL-10 levels are higher in helminth infection since these parasites have mechanisms of evading the immune system to ensure long-term survival within the host [46]. IL-10 and FoxP3 were increased in the *A. lumbricoides* infection and coinfection groups, suggesting that IL-10 and FoxP3 are upregulated by *A. lumbricoides*. Dual infection stimulation was done to elucidate coinfection scenarios and to determine if there is an effect in the up or downregulation of Th1, Th2, and regulatory cytokines. Regulatory cytokines are high during *A. lumbricoides* infection and also in cases of TB/helminth coinfection compared to TB infection. This may be due to the downmodulation of the immune response to TB.

The present study demonstrated a typical TB response characterized by an increase in inflammatory cytokines such as IFN-γ, TNF-α, IL-2, and IL-17. However, we did not use costimulatory molecules such as anti-CD 28 or anti-Cd49d to enhance the stimulation of the Jurkat cells, since they do not possess antigen-presenting properties. Therefore, the Jurkat cell response may be suboptimal, due to the exclusion of immuno co-stimulatory molecules, which is a limitation of this study. 

## 5. Study Limitations

The current study was limited by the small sample sizes in the TB-infected and TB plus helminth-coinfected participant groups. However, the limited analysis mirrored what was found in in vitro experiments, which showed higher pro-inflammatory/Th1 in the TB-infected group and lower in the coinfected group. An unexpected, possibly spurious finding was that some levels of cytokine gene transcription levels levels were highest among the uninfected controls, such as perforin, NFATC-2, TGF-β, and IL-10. This may be because the uninfected controls were only screened for helminths and TB in the laboratory, while they could possibly be exposed to other bacterial, viral, or other immune-activating factors that could not be detected during the questionnaire administration that was used in the main study. The demographic profile of the participants may attest to the possibility of other immune-activating environmental factors [29]. In addition, as alluded to above, the cell culture experiments in the in vitro studies did not include co-stimulatory molecules to properly represent the in vivo antigen presentation and processing.Therefore, these results were suboptimal, despite the fact that the main responses typically depicted TB (Th1/inflammatory) and helminth (Th2/Treg) profiles. 

As noted above, there are additional tests, such as flow cytometry and tunnel assays that could be performed to validate the increase in granzyme B and perforin.

The gene transcription levels of cytokines in stimulated and unstimulated Jurkat and THP-1 cell lines are not directly correlated with its production. Therefore, to validate the gene transcription levels results, further tests such as ELISA needed to be performed.

## 6. Conclusions

The in vitro findings suggest that pro-inflammatory Th1 responses are increased in TB infection and reduced in cases of coinfection. The study also determined that anti-inflammatory Th2 and regulatory cytokines are increased during single helminth infection and in TB and helminth coinfection. The ex vivo data, although limited by the sample size, also supported the hypothesis that TB increases Th1 immune responses and responses to helminths involve strong Th2 and regulatory cytokines.

## Figures and Tables

**Figure 1 microorganisms-11-01846-f001:**
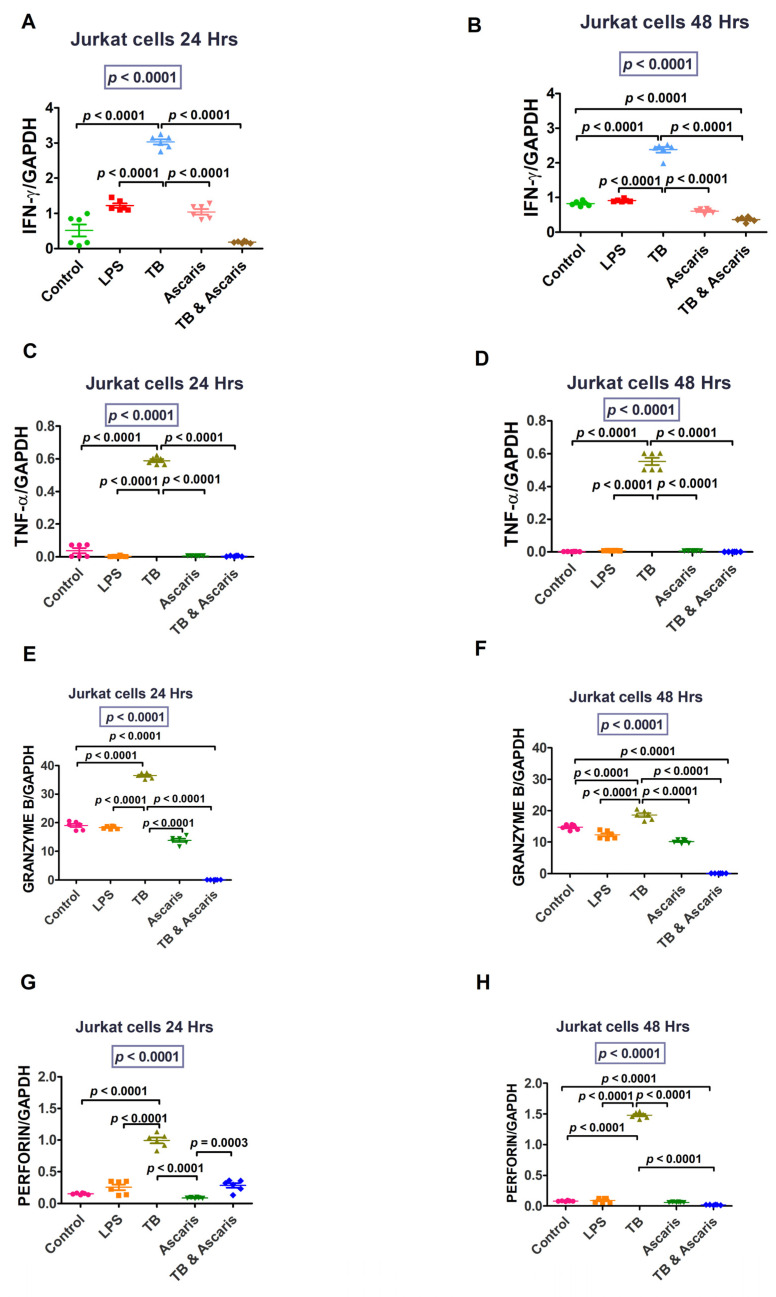
Transcription levels data for IFN-γ, TNF-α, granzyme B, and perforin are shown for Jurkat cells at 24 and 48 h time points. The unstimulated control cells and LPS-stimulated cells were negative and positive controls, respectively. (**A**) IFN-γ levels for Jurkat cells stimulated for 24 h, (**B**) IFN-γ levels for Jurkat cells stimulated for 48 h, (**C**) TNF-α levels for Jurkat cells stimulated for 24 h, (**D**) TNF-α levels for Jurkat cells stimulated for 48 h, (**E**) granzyme B levels for Jurkat cells stimulated for 24 h, (**F**) granzyme B levels for Jurkat cells stimulated for 48 h, (**G**) perforin levels for Jurkat cells stimulated for 24 h, and (**H**) perforin levels for Jurkat cells stimulated for 48 h.

**Figure 2 microorganisms-11-01846-f002:**
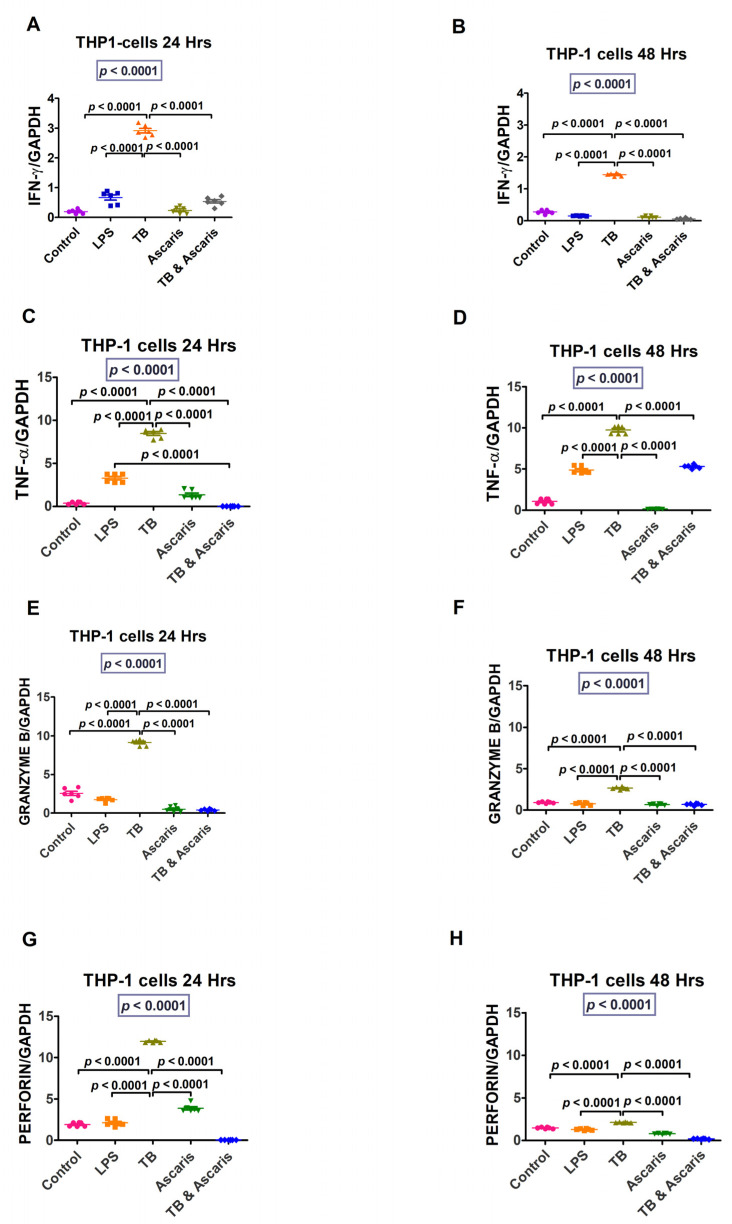
Transcription level data for IFN-γ, TNF-α, granzyme B, and perforin are shown for THP-1 cells at 24 and 48 h time points. The unstimulated control cells and LPS-stimulated cells were negative and positive controls, respectively. (**A**) IFN-γ levels for THP-1 cells stimulated for 24 h, (**B**) IFN-γ levels for THP-1 cells stimulated for 48 h, (**C**) TNF-α levels for THP-1 cells stimulated for 24 h, (**D**) TNF-α levels for THP-1 cells stimulated for 48 h, (**E**) granzyme B levels for THP-1 cells stimulated for 24 h, (**F**) granzyme B levels for THP-1 cells stimulated for 48 h, (**G**) perforin levels for THP-1 cells stimulated for 24 h, and (**H**) perforin levels for THP-1 cells stimulated for 48 h.

**Figure 3 microorganisms-11-01846-f003:**
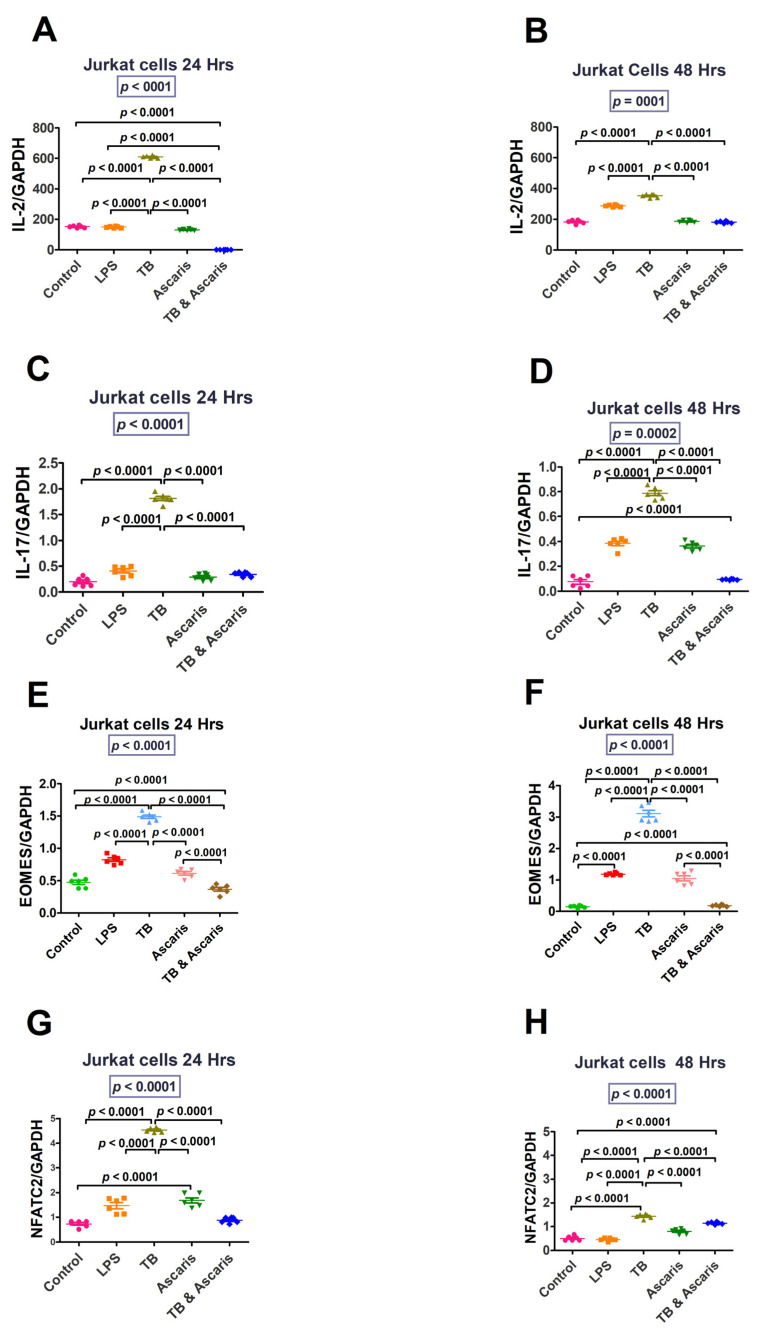
Jurkat cell line responses for Th1/pro-inflammatory (IL-2 and IL-17) and transcription factors (Eomes and NFATC2) at 24 and 48 h time points. The unstimulated control cells and LPS-stimulated cells were negative and positive controls, respectively. (**A**) IL-2 levels for Jurkat cells stimulated for 24 h, (**B**) IL-2 levels for Jurkat cells stimulated for 48 h, (**C**) IL-17 levels for Jurkat cells stimulated for 24 h, (**D**) IL-17 levels for Jurkat cells stimulated for 48 h, (**E**) EOMES levels for Jurkat cells stimulated for 24 h, (**F**) EOMES—transcription levels for Jurkat cells stimulated for 48 h, (**G**) NFATC2 levels for Jurkat cells stimulated for 24 h, and (**H**) NFATC2 levels for Jurkat cells stimulated for 48 h.

**Figure 4 microorganisms-11-01846-f004:**
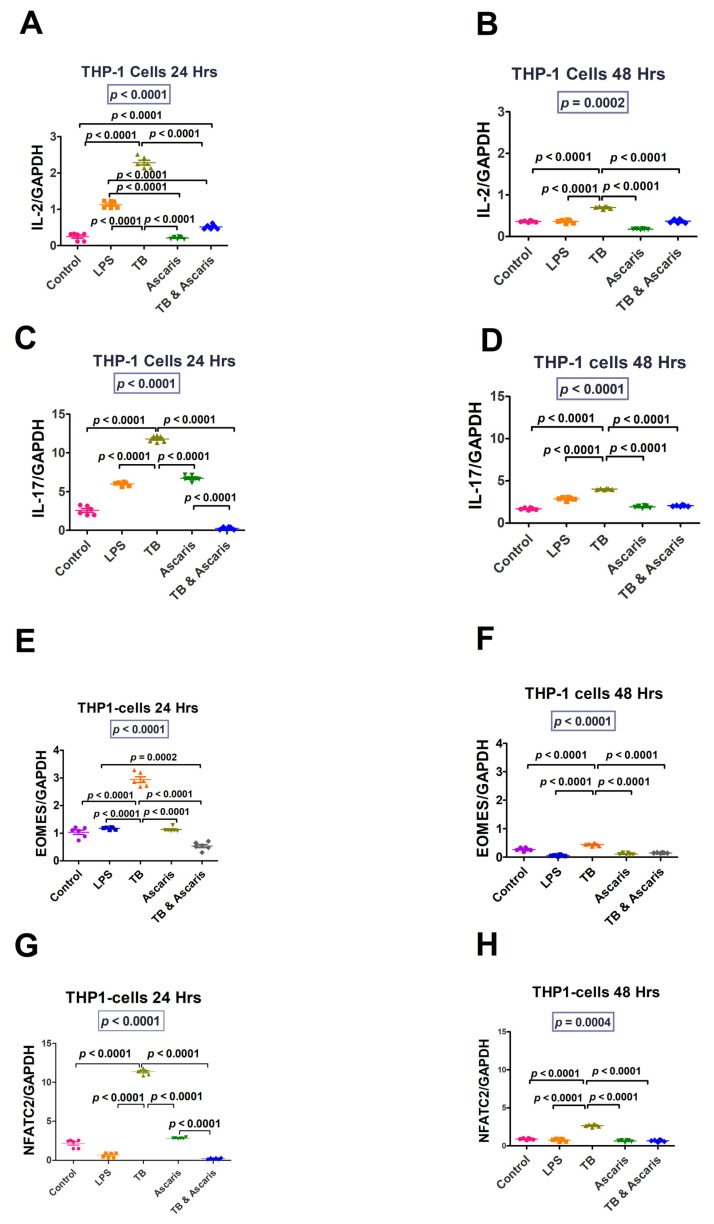
THP-1 cell line responses for Th1/pro-inflammatory (IL-2 and IL-17) and transcription factors (Eomes and NFATC2) at 24 and 48 h time points. The unstimulated control cells and LPS-stimulated cells were negative and positive controls, respectively. (**A**) IL-2 levels for THP-1 cells stimulated for 24 h, (**B**) IL-2 levels for THP-1 cells stimulated for 48 h, (**C**) IL-17 levels for THP-1 cells stimulated for 24 h, (**D**) IL-17—transcription levels for THP-1 cells stimulated for 48 h, (**E**) EOMES levels for THP-1 cells stimulated for 24 h, (**F**) EOMES levels for THP-1 cells stimulated for 48 h, (**G**) NFATC2 levels for THP-1 cells stimulated for 24 h, and (**H**) NFATC2 levels for THP-1 cells stimulated for 48 h.

**Figure 5 microorganisms-11-01846-f005:**
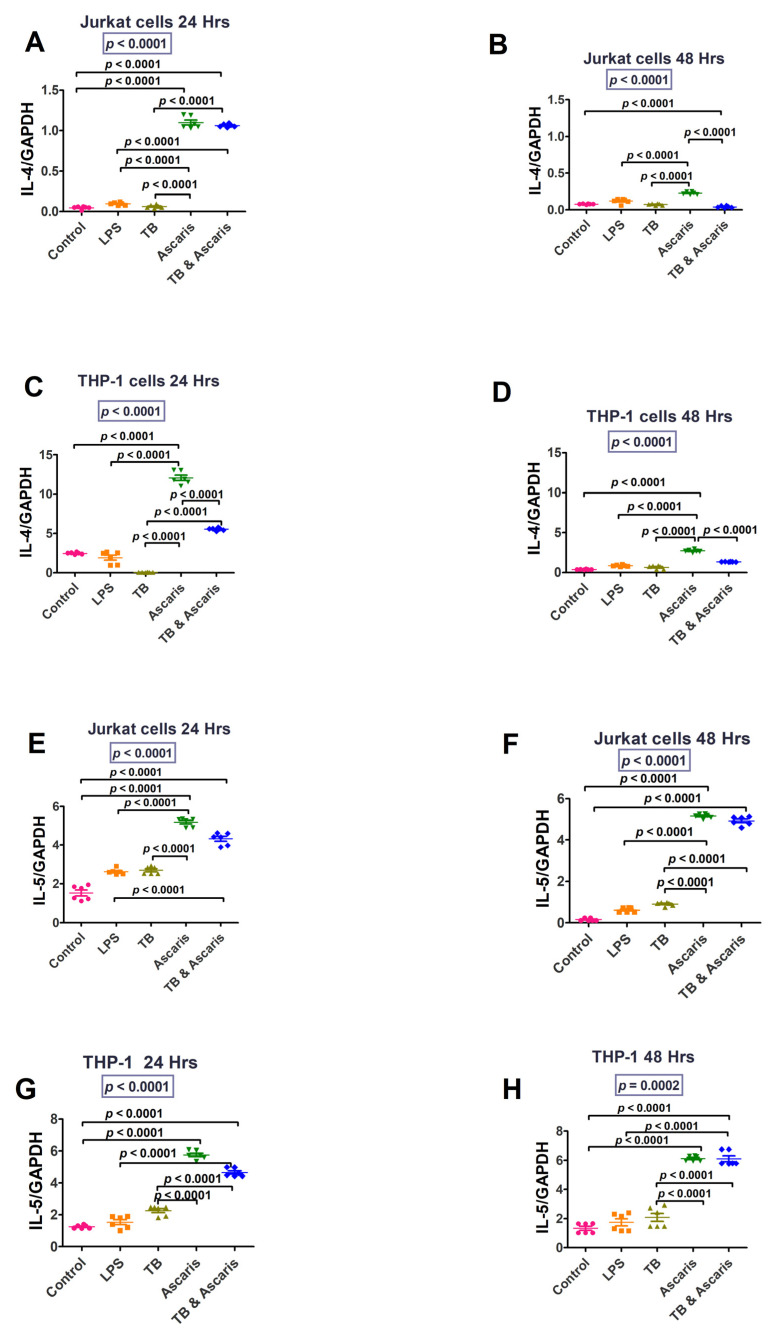
IL-4 and IL-5 responses in TB, *A. lumbricoides*, and dually stimulated Jurkat and THP-1 cells at 24 and 48 h time points. (**A**) IL-4 levels for Jurkat cells stimulated for 24 h, (**B**) IL-4 levels for Jurkat cells stimulated for 48 h, (**C**) IL-4—levels for THP-1 cells stimulated for 24 h, (**D**) IL-4 levels for THP-1 cells stimulated for 48 h, (**E**) IL-5 levels for Jurkat cells stimulated for 24 h, (**F**) IL-5 levels for Jurkat cells stimulated for 48 h, (**G**) IL-5 levels for THP-1 cells stimulated for 24 h, and (**H**) IL-5—transcription levels for THP-1 cells stimulated for 48 h.

**Figure 6 microorganisms-11-01846-f006:**
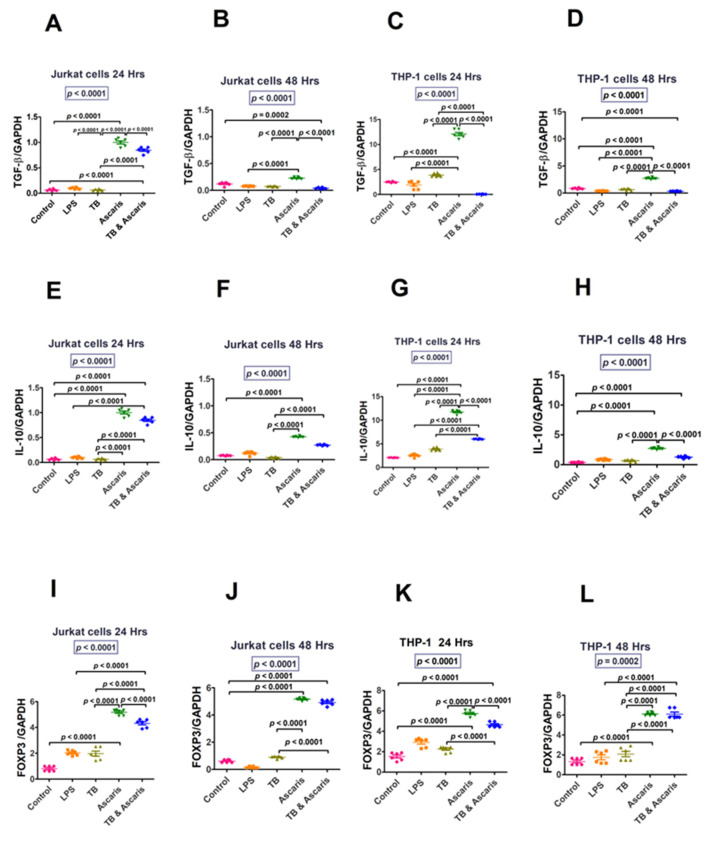
Jurkat and THP1 cell responses for regulatory cytokines (TGF-β, IL-10, and FoxP3) at 24 and 48 h time points. The unstimulated control and LPS-stimulated cells were used as negative and positive controls, respectively. (**A**) TGF-β levels for Jurkat cells stimulated for 24 h, (**B**) TGF-β levels for Jurkat cells stimulated for 48 h, (**C**) TGF-β levels for THP-1 cells stimulated for 24 h, (**D**) TGF-β levels for THP-1 cells stimulated for 48 h, (**E**) IL-10 levels for Jurkat cells stimulated for 24 h, (**F**) IL-10 levels for Jurkat cells stimulated for 48 h, (**G**) IL-10 levels for THP-1 cells stimulated for 24 h, (**H**) IL-10 levels for THP-1 cells stimulated for 48 h, (**I**) FoxP3 levels for Jurkat cells stimulated for 24 h, (**J**) FoxP3 levels for Jurkat cells stimulated for 48 h, (**K**) FoxP3 levels for THP-1 cells stimulated for 24 h, (**L**) FoxP3 levels for THP-1 cells stimulated for 48 h.

**Figure 7 microorganisms-11-01846-f007:**
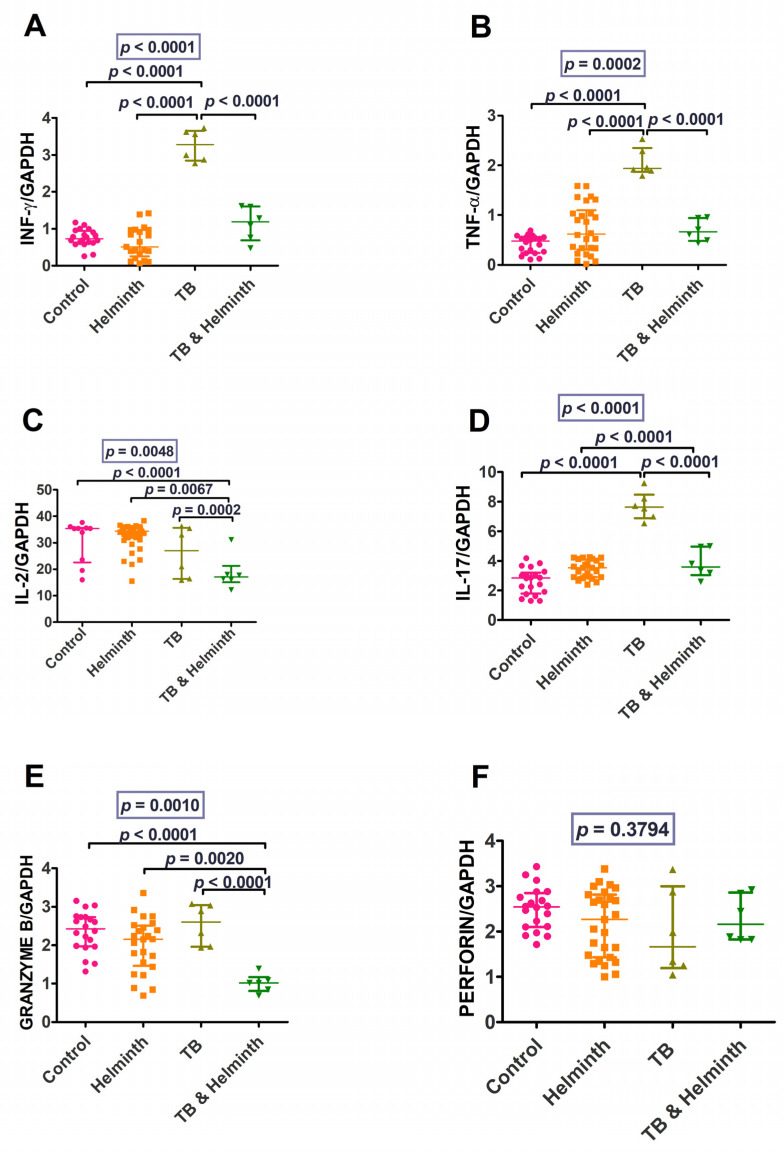
IFN-γ, TNF-α, IL2, IL-17, perforin, and granzyme B ex vivo gene transcription levels data. (**A**) IFN-γ levels, (**B**) TNF-α levels, (**C**) IL-2 levels, (**D**) IL-17 levels, (**E**) granzyme B levels, and (**F**) perforin levels.

**Figure 8 microorganisms-11-01846-f008:**
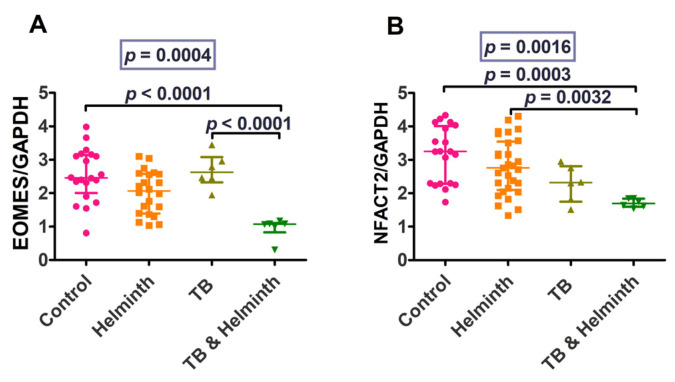
Eomes and NFATC2 gene transcription levels values. (**A**) Eomes levels, and (**B**) NFATC2 levels.

**Figure 9 microorganisms-11-01846-f009:**
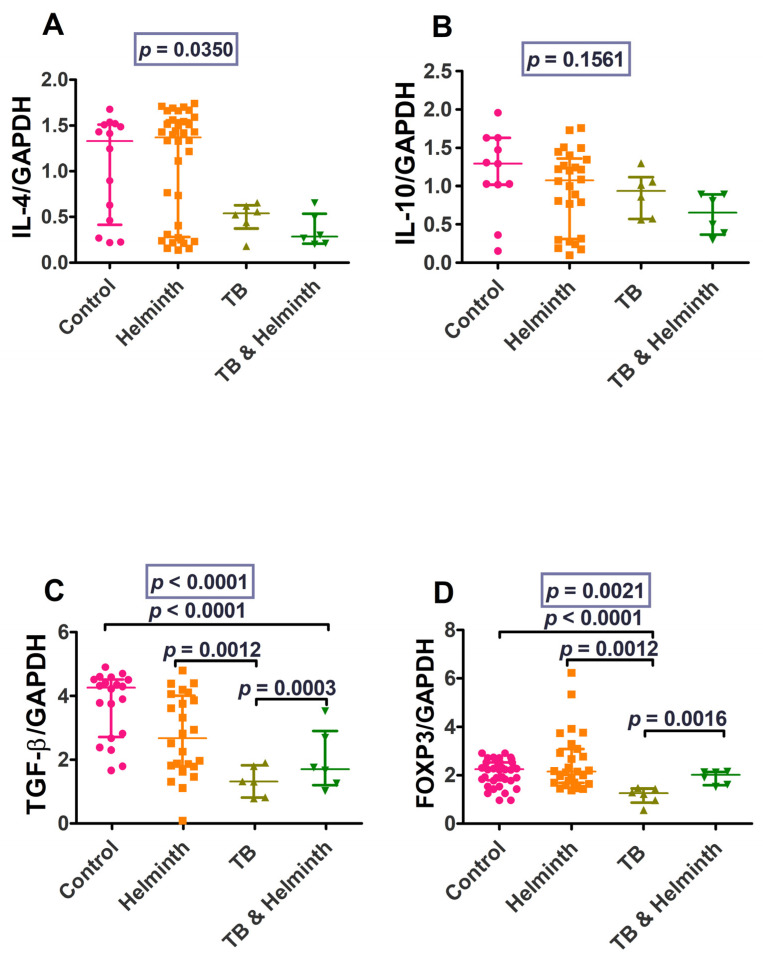
IL-4, IL-10, TGF-β, and FoxP3 gene transcription levels values. (**A**) IL-4 levels, (**B**) IL-10 levels, (**C**) TGF-β levels, and (**D**) FoxP3 levels.

**Table 1 microorganisms-11-01846-t001:** List of abbreviations and acronyms used in the paper.

Abbreviation	Definition
ATCC	American Type Culture Collection
*A. lumbricoides*	*Ascaris lumbricoides*
BCG	Bacille Calmette-Guerin
Eomes	Eomesodermin
ESP	Excretory-secretory protein
FoxP3	Forkhead box P3
GAPDH	Glyceraldehyde 3 diphosphate dehydrogenase
IFN-γ	Interferon-gamma
IL	Interleukin
LPS	Lipopolysaccharide
MHC	Major histocompatibility complex
Mtb	*Mycobacterium tuberculosis*
Nb	*Nippostrongylus brasiliensis*
NFATC2	Nuclear factor of activated T-cells
OADC	Oleic acid albumin dextrose catalase enrichment
RT-qPCR	Real-time quantitative polymerase chain reaction
SA	South Africa
TGF-β	Transforming growth factor-beta
Th1	T-helper type 1
Th2	T-helper type 2
TNF-α	Tumour necrosis factor-alpha
Treg	Regulatory T cells

**Table 2 microorganisms-11-01846-t002:** FAM-labelled cytokine probe mix purchased from Thermofisher Scientific and their corresponding catalogue number. The cytokines and transcription factors were chosen based on the immune response—[Th-1 and Th-17 cytokines and transcription factors (INF-γ, TNF-α), IL2, IL17, Granzyme B, perforin, NFATC2 and Eomes), Th-2 (IL-4, IL5), Regulatory cytokines and transcription factors (TGF-β, IL-10 and FoxP3)].

Cytokine Gene	Thermofisher Catalogue Number
Glyceraldehyde 3-phosphate dehydrogenase (GAPDH) (housekeeping gene)	Hs99999905_m1
Interferon-gamma (INF-γ)	Hs00989291_m1
Tumour necrosis factor-alpha (TNF-α)	Hs00174128_m1
Granzyme B	Hs00188051_m1
Perforin	Hs00169473_m1
Interleukin-2 (IL-2)	Hs00174114_m1
Interleukin-17 (IL-17)	Hs01056316_m1
Nuclear factor of activated T-cells 2 (NFATC2)	Hs00905451_m1
Eomesodermin (Eomes)	Hs00172872_m1
Interleukin-4 (IL-4)	Hs00174122_m1
Interleukin-5 (IL-5)	Hs99999031_m1
Interleukin-10 (IL-10)	Hs00961622_m1
Transforming growth factor beta (TGF-β)	Hs00234244_ml
Forkhead box P3 (FoxP3)	Hs01085834_m1

## Data Availability

The data collected for the current study are available on request from the corresponding author.

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
