# Peer review of "Cytokine Responses during Mycobacterium tuberculosis H37Rv and Ascaris lumbricoides Costimulation Using Human THP-1 and Jurkat Cells, and a Pilot Human Tuberculosis and Helminth Coinfection Study"

_microorganisms, 2023, doi:10.3390/microorganisms11071846_

Round 1
Reviewer 1 Report
General : The manuscript is well written and the topic has great significance.
Abstract: Line 24- modify the statement. There are several studies that show the effect of helminth infection on BCG vaccination and TB but mechanisms remains unclear.
Method: Line 116 and 117: It appears that Jurkat cells were stimulated with the different antigens directly. How effective are Jurkat cells in antigen processing and presentation?
Results:
Figure 1E and 1F: Explain why Jurkat cells produce low level of cytokines for LPS which is supposed to be a positive control.
Line 246: ... 35 were helminth infected.. What type of helminths? How many of these had Ascaris?
Figure 5A-D: Was there a significant difference between the group with TB alone and group with TB & helminth. This is the only comparison that could tell us the impact of helminth infection.
Figure 7 A-D: Was there a significant difference between TB vs TB plus helminth groups?
Discussion
Line 297 and 298: ...Th2 and regulatory cytokines were increased in dual infection. Figure 7 does not appear to support this conclusion.
Author Response
Comments from the reviewer: The manuscript is well-written, and the topic has great significance.
Abstract:
Line 24- modify the statement. There are several studies that show the effect of helminth infection on BCG vaccination and TB, but mechanisms remain unclear.
The statement has been modified. Please refer to lines 28 – 29.
Method:
Line 116 and 117: It appears that Jurkat cells were stimulated with the different antigens directly. How effective are Jurkat cells in antigen processing and presentation?
Jurkat cells were stimulated directly. This is a limitation for the study because the Jurkat cell response may be suboptimal because costimulatory molecules were absent because these cells do not present antigens. However, TB has been shown to have superantigen properties which do not require antigen-presenting cells. Please refer to lines 557 – 563, 578 -580.
Results:
Figure 1E and 1F: Explain why Jurkat cells produce low level of cytokines for LPS which is supposed to be a positive control.
Jurkat cell response may be suboptimal due to exclusion of anti CD28 which is a costimulatory molecule. Please refer to lines 557 – 563, 578 -580.
Line 246: ... 35 were helminth infected. What type of helminths? How many of these had Ascaris?
The infecting helminth species have been listed. Please refer to lines 196-199, 416.- 419.
Figure 5A-D: Was there a significant difference between the group with TB alone and group with TB & helminth. This is the only comparison that could tell us the impact of helminth infection.
Apologies for initially not including these findings. There were significant differences between the TB alone and TB & helminth for IFN-γ, TNF- α, IL-2, and IL-17.
Figure 7 A-D: Was there a significant difference between TB vs TB plus helminth groups?
There were no significant differences between TB and TB plus helminth for IL-4 and IL-10 however, these were noted for TGF-β and transcription factor FoxP3.
Discussion
Line 297 and 298: ...Th2 and regulatory cytokines were increased in dual infection. Figure 7 does not appear to support this conclusion.
The statement has been amended.

Reviewer 2 Report
General description
Authors took a nice survey to Explorer interactions between two different classes of pathogens and immune system. Helminths are multicellular organisms that shift immune response towards Th2 phenotype which may affect susceptibility to Th1 prone pathogens and vaccine efficiency. The purpose of the described experiments is well justified and the results are presented very clearly. The figures are informative and leave no doubts for an interpretation. Summing up, the concept and the paper quality are very high
Drawbacks
Despite high quality of the paper there are some major and minor remarks that need to be improved.
Major drawbacks
The conclusions and the results are achieved based on QT-PCR technique. The description of the methodology is insufficient.
· Authors did not perform DNAs treatment of RNA samples. Genomic DNA in RNA samples is obvious and may lead to false results. Authors used commercially available sets of primers/probes to analyze expression of particular cDNA. Authors should provide information if those sets amplify genomic cDNA or perform experiments proving that the amplified products were not amplified on genomic DNA matrix.
· However Authors mentioned about statistical analyses and use of particular statistical tests, they did not specify on what bases they achieved values of expression of particular cDNAs (level of particular cDNA/ level to cDNA encoding GAPDH). This is usually done using Livak and Schmittgen or Pfaffl methods, however other approaches are also acceptable.
· Authors also depicted serial dilution of pooled cDNA matrix from 1000 ng to 1 ng. What are the units? Were those values expressen in 1000 ng/µl or 1000 ng/reaction. What was the buffer to perform the dilutions. Was it water? If so, increased expression may be observed in higher dilution since RT mixture components may inhibit the reaction. I observed such cases number of times.
Authors should put more effort in justifying use the models: THP-1 and Jurkat cells, especially the latter one. THP-1 are monocytes that respond to antigens. Jurkat cells are lymphosytes that respond to antigens upon presentation. Authors should discuss the ability of Jurkat cells to reply to antigens without presentation.
Authors should specify what helminths were harbored by the patients.
Authors used ES of Ascaris lumbricoides, more detailed description of collecting antigens is required, especially the information if endotoxins were measures.
Minor remarks:
In the discussion there is information regarding expression of regulatory cytokines: IL-10 and FoxP3. FoxP3 is not a cytokine but a transcription factor. This should be changed.
Authors should give Cat No. of LPS they used. There are number of LPSs.
Line 353: “IL5” should be changed to “IL-5”
When describing matrix dilution, Authors use “x” instead of the appropriate symbol “×”. Similar mistake occurs when Authors describe units “5 ug/ml”… again the appropriate symbol for “micro” value exists – “µ”
Author Response
Comments from the reviewer: Authors took a nice survey to Explorer interactions between two different classes of pathogens and the immune system. Helminths are multicellular organisms that shift the immune response towards the Th2 phenotype, which may affect susceptibility to Th1-prone pathogens and vaccine efficiency. The purpose of the described experiments is well justified, and the results are presented very clearly. The figures are informative and leave no doubts for interpretation. Summing up, the concept and the paper quality are very high
Drawbacks
Despite the high quality of the paper, there are some major and minor remarks that need to be improved.
Major drawbacks
The conclusions and the results are achieved based on QT-PCR technique. The description of the methodology is insufficient.
- Authors did not perform DNAs treatment of RNA samples. Genomic DNA in RNA samples is obvious and may lead to false results. Authors used commercially available sets of primers/probes to analyze expression of particular cDNA. Authors should provide information if those sets amplify genomic cDNA or perform experiments proving that the amplified products were not amplified on genomic DNA matrix.
The DNase treatment was performed. Please refer to lines 152 – 159.
- However Authors mentioned about statistical analyses and use of particular statistical tests, they did not specify on what bases they achieved values of expression of particular cDNAs (level of particular cDNA/ level to cDNA encoding GAPDH). This is usually done using Livak and Schmittgen or Pfaffl methods, however other approaches are also acceptable.
The standard curve method was used to calculate gene expression, whereby the value of the target gene was divided by the GAPDH value. Values were expressed as medians. Please refer to lines 211 – 213, 215 - 219.
- Authors also depicted serial dilution of pooled cDNA matrix from 1000 ng to 1 ng. What are the units? Were those values expressed in 1000 ng/µl or 1000 ng/reaction? What was the buffer to perform the dilutions? Was it water? If so, increased expression may be observed in higher dilution since RT mixture components may inhibit the reaction. I observed such cases a number of times.
The values of pooled cDNA were expressed as 1000ng/µl. Please refer to line 185.
The buffer used was RNase water. To account for inhibited reaction, if the amplification was not linear, then those samples were excluded.
Authors should put more effort in justifying use the models: THP-1 and Jurkat cells, especially the latter one. THP-1 are monocytes that respond to antigens. Jurkat cells are lymphocytes that respond to antigens upon presentation. Authors should discuss the ability of Jurkat cells to reply to antigens without presentation.
Jurkat cells were stimulated directly. This is a limitation for the study because the Jurkat cell response may be suboptimal because costimulatory molecules were absent and these cells do not present antigens. However, TB has been shown to have superantigen properties which do not require antigen-presenting cells. Please refer to lines 557 – 563, 578 -580.
Authors should specify what helminths were harbored by the patients.
The infecting intestinal parasites were listed. Please refer to lines 196-199, 416.- 419.
Authors used ES of Ascaris lumbricoides, more detailed description of collecting antigens is required, especially the information if endotoxins were measures.
The method has been explained. Please refer to lines 118 – 127.
Minor remarks:
In the discussion, there is information regarding the expression of regulatory cytokines: IL-10 and FoxP3. FoxP3 is not a cytokine but a transcription factor. This should be changed.
This has been changed. Please refer to line number 540.
Authors should give Cat No. of LPS they used. There are number of LPSs.
The catalogue number has been added. Please refer to line 138.
Line 353: “IL5” should be changed to “IL-5”
This has been amended. Please refer to line 535.
When describing matrix dilution, Authors use “x” instead of the appropriate symbol “×”. A similar mistake occurs when Authors describe units “5 ug/ml”… again the appropriate symbol for “micro” value exists – “µ”
The symbols have been amended. Please refer to lines number 133,141, 142, 143, 144.

Reviewer 3 Report
The paper by Bhengu et al entitled “Cytokine responses during Mycobacterium tuberculosis H37Rv and Ascaris lumbricoides co-infection using human THP-1 and Jurkart cell, and a pilot human tuberculosis and helminth co-infection study” describe the expression of several cytokines in cell lines stimulated with M. tuberculosis or/and Ascaris lumbricoides. In addition, the authors evaluated the expression of the same genes used in vitro in a pilot study in humans. The choice of the two microrganisms is well described in the Introduction section, as well as the immune response that these two pathogens trigger.
One critical issue which is not easy to understand pertains the methodology used. Specifically, the relevance of stimulating a T cell line with antigens from M. tuberculosis and A. lumbricoides. Since there are no APC in the culture the response is likely to PAMPs present in the antigen preparations. The use of this cell line should be clarified.
Figure 1:
- Page 6 line 202: the authors report a significant increase in IFNg in the THP-1 and Jurkat cell lines after M. tuberculosis stimulation. Looking at the Figure 1 A also the Jurkat cell line stimulated with Ascaris group show an increase expression of IFNg when compared to the unstimulated control and the co-infected condition. The same situation is clearly observed in Figure 1B and in the Figure 1C. If the data is not statistically significant, it should be clarified in the figures. The authors only show p<0.0001.
- Page 6 line 203: The authors described an increase in TNF in TB-stimulated compared to the Ascaris and the co-infected groups. Figure 1 G and Figure 1 H show also an increase in TNF expression in the group of THP-1 cell line stimulated with Ascaris and in the co-infected group, respectively. Statistical significance should be stated.
- Page 6-7, lines 205-206: Regarding the Perforin, the authors report increase in its expression in the TB-stimulated cells when compared to the co-stimulated group. Figure 1 O show also an increase of this protein in the THP-1 cells stimulated with Ascaris alone.
- The authors in the section 2 Materials and Methods, specifically in page 3, line 117 explain that they use unstimulated group for each cell line as a negative control. Since it is a negative control why this group display an increased expression of IFNg, granzyme and perforin genes?
Overall, the authors should describe better section of Results related to the Figure 1, reporting not only the most significant results, but also the other changes showed in each graph.
Furthermore, the authors should improve disposition of the graph to make them more readable (adding a title with the subject analysed and dividing the two cell lines used in two different subparagraph).
Page 7 line 206-207: the authors should move the acronym in the figure legend or use a specific section for acronyms.
Figure 2:
The description of the graphs should be coherent with the Figure 1, describing the difference observed between the different groups instead of reporting only the p value.
In addition, the graphs should have the same size, the size of the font used should be the same for each graph.
Page 7 line 216: the authors should move the acronym in the figure legend or use a specific section for acronyms.
Figure 3:
- Page 8 line 225: it is not clear what the authors mean with this sentence. Indeed, in the Figure 1 H, shows an increased expression of IL-5 in both stimulated and co-stimulated cells with Ascaris when compared with the group stimulated with M.tuberculosis.
Figure 4:
- Page 9, lines 227-230: The authors report an increase in expression of TGFb and IL-10 in the Ascaris stimulated group and in the co-stimulated group in both cell lines at 24 and only in the Ascaris stimulated group at 48h. But looking at the Figure 4 F and Figure 4 H it is possible observed that also the co-stimulated group show an increase in IL-10 for both cell lines.
Page 10 lines 236-239: The authors should use a more formal English and describe correctly the graph not only report the p value.
Page 10 lines 239-240: the authors should move the acronym in the figure legend or use a specific section for acronyms.
Figure 6:
- Page 10, line 259: The authors should tone down when they describe a higher increase in IL-4 and IL-10 in the helmint groups, since the graph display a high variability.
Page 14 lines 297-298: the authors should rephrase the sentence, since they showed that the Th2 and the regulatory cytokines in both cells lines were increased in the Ascaris stimulated group. In addition, in the humans those response were no different among the groups.
Page 14 lines 316-318: the authors should tone down, to validate that the increase in granzyme and perforin showed by the cells after 24h from the stimulation, they should perform a proper experiment like TUNEL assay, or using flow cytometry.
Page 14, lines 332-335: the author should clarify which is the role of IL-17 in their model.
Page 15, lines 358: FoxP3 is not a cytokine is a transcription factor !
The authors should readjust the graphs and the legend of the Figures to be consistent among them (size of the graph, presentation of the data). In addition, the authors should improve the readability of the “Part 1: in vitro study”, such as dividing the paragraph in two subparagraphs one for the THP-1 cell line and the other for the Jurkart cell line.
All the acronyms used in the article should be moved in the figure legend or in a specific section for acronyms.
In this study the authors evaluate only the gene expression of cytokines in stimulated and unstimulated THP-1 and Jurkart cell lines. Since the expression of a protein is not directly correlated with its production, it will be important to perform an ELISA to quantify the concentration of those cytokines since the amount of mRNA produce doesn’t correlate with the real concentration of protein released.
The results needs rewriting, since the authors took too much care about the p value, without described the graph and explain what they observed. In addition, the English used in the entire paper should be coherent.
Furthermore, the authors should re-write the discussion section since it poorly describes, and it is not so well connected with the results that the authors obtained.
The english language is fine.
Author Response
Reviewer #3 :
Comments from the reviewer:
The paper by Bhengu et al entitled “Cytokine responses during Mycobacterium tuberculosis H37Rv and Ascaris lumbricoides co-infection using human THP-1 and Jurkat cell, and a pilot human tuberculosis and helminth co-infection study” describes the expression of several cytokines in cell lines stimulated with M. tuberculosis or/and Ascaris lumbricoides. In addition, the authors evaluated the expression of the same genes used in vitro in a pilot study in humans. The choice of the two microorganisms is well described in the Introduction section, as well as the immune response that these two pathogens trigger.
One critical issue which is not easy to understand pertains the methodology used. Specifically, the relevance of stimulating a T cell line with antigens from M. tuberculosis and A. lumbricoides. Since there are no APCs in the culture, the response is likely to PAMPs present in the antigen preparations. The use of this cell line should be clarified.
Figure 1:
- Page 6, line 202: the authors report a significant increase in IFN-γ in the THP-1 and Jurkat cell lines after M. tuberculosis stimulation. Looking at Figure 1 A also, the Jurkat cell line stimulated with the Ascaris group shows an increase expression of IFN-γ when compared to the unstimulated control and the co-infected condition. The same situation is clearly observed in Figure 1B and in Figure 1C. If the data is not statistically significant, it should be clarified in the figures. The authors only show p<0.0001.
The results have been explained extensively. Please refer to the results section.
- Page 6 line 203: The authors described an increase in TNF in TB-stimulated compared to the Ascaris and the co-infected groups. Figure 1 G and Figure 1 H show also an increase in TNF expression in the group of THP-1 cell line stimulated with Ascaris and in the co-infected group, respectively. Statistical significance should be stated.
The results have been explained. Please refer to lines 269 – 286.
- Page 6-7, lines 205-206: Regarding the Perforin, the authors report increase in its expression in the TB-stimulated cells when compared to the co-stimulated group. Figure 1 O show also an increase of this protein in the THP-1 cells stimulated with Ascaris alone.
The results have been described more. Please refer to lines 284 – 286.
- The authors in the section 2 Materials and Methods, specifically in page 3, line 117 explain that they use unstimulated group for each cell line as a negative control. Since it is a negative control why this group display an increased expression of IFNg, granzyme and perforin genes?
Overall, the authors should describe better section of Results related to the Figure 1, reporting not only the most significant results, but also the other changes showed in each graph.
Furthermore, the authors should improve disposition of the graph to make them more readable (adding a title with the subject analysed and dividing the two cell lines used in two different subparagraph).
The results have been described more. Please refer to lines 244 – 262 and lines 269 – 286.
Page 7 line 206-207: the authors should move the acronym in the figure legend or use a specific section for acronyms.
Acronyms have been moved to their section. Please refer to lines 99 – 102.
Figure 2:
The description of the graphs should be coherent with the Figure 1, describing the difference observed between the different groups instead of reporting only the p value.
In addition, the graphs should have the same size, the size of the font used should be the same for each graph.
The results have been explained and graphs reformatted. Please refer to results section.
Page 7 line 216: the authors should move the acronym in the figure legend or use a specific section for acronyms.
Acronyms have been moved to their section. Please refer to lines 99 – 102.
Figure 3:
- Page 8 line 225: it is not clear what the authors mean with this sentence. Indeed, in the Figure 1 H, shows an increased expression of IL-5 in both stimulated and co-stimulated cells with Ascaris when compared with the group stimulated with M.tuberculosis.
The results have been explained more. Please refer to 373 -379.
Figure 4:
- Page 9, lines 227-230: The authors report an increase in expression of TGFb and IL-10 in the Ascaris stimulated group and in the co-stimulated group in both cell lines at 24 and only in the Ascaris stimulated group at 48h. But looking at the Figure 4 F and Figure 4 H it is possible observed that also the co-stimulated group show an increase in IL-10 for both cell lines.
The results have been explained more. Please refer to lines 400 – 406.
Page 10 lines 236-239: The authors should use more formal English and describe correctly the graph not only report the p value.
The results have been amended Please refer to lines 400 – 406.
Page 10 lines 239-240: the authors should move the acronym in the figure legend or use a specific section for acronyms.
Acronyms have been moved to their section. Please refer to lines 99 – 102.
Figure 6:
- Page 10, line 259: The authors should tone down when they describe a higher increase in IL-4 and IL-10 in the helmint groups, since the graph display a high variability.
The explanation has been amended. Please refer to lines 431 – 433.
Page 14 lines 297-298: the authors should rephrase the sentence, since they showed that the Th2 and the regulatory cytokines in both cells lines were increased in the Ascaris stimulated group. In addition, in the humans those response were no different among the groups.
The sentence has been rephrased. Please refer to lines 476 – 478.
Page 14 lines 316-318: the authors should tone down, to validate that the increase in granzyme and perforin showed by the cells after 24h from the stimulation, they should perform a proper experiment like TUNEL assay, or using flow cytometry.
Please refer to lines 498 – 500 and lines 581 – 582.
Page 14, lines 332-335: the author should clarify which is the role of IL-17 in their model.
Please refer to lines 508 -514.
Page 15, lines 358: FoxP3 is not a cytokine is a transcription factor !
The statement has been amended. Please refer to line 540.
The authors should readjust the graphs and the legend of the Figures to be consistent among them (size of the graph, presentation of the data). In addition, the authors should improve the readability of the “Part 1: in vitro study”, such as dividing the paragraph in two subparagraphs one for the THP-1 cell line and the other for the Jurkart cell line.
Graphs have been amended.
All the acronyms used in the article should be moved in the figure legend or in a specific section for acronyms.
Acronyms have been moved to their section. Please refer to lines 99 – 102.
In this study the authors evaluate only the gene expression of cytokines in stimulated and unstimulated THP-1 and Jurkart cell lines. Since the expression of a protein is not directly correlated with its production, it will be important to perform an ELISA to quantify the concentration of those cytokines since the amount of mRNA produce doesn’t correlate with the real concentration of protein released.
The statement has been added to the limitation section. Please refer to lines 583 -585.
The results needs rewriting, since the authors took too much care about the p value, without described the graph and explain what they observed. In addition, the English used in the entire paper should be coherent.
The results have been amended. Please refer to the results section.
Furthermore, the authors should re-write the discussion section since it poorly describes, and it is not so well connected with the results that the authors obtained.
The discussion has been amended

Round 2
Reviewer 2 Report
The MS was significantly improved.
Author Response
Reviewer's comments - The MS was significantly improved.
Thank you, reviewer.
Reviewer 3 Report
The paper by Bhengu and colleagues maintained some major issues after the revision that still require the author’s attention, specifically in what regards the interpretation and description of the data and the scientific principles described. As an example, description of figure 1 is not clear at all, with graphs showing something and the authors describing something else (e.g. no significance in IFNg between TB and Ascaris when the graph shows otherwise; lines 251-253 the authors contradict themselves; the use of tunnel and flow cytometry to confirm the data is not clear; in multiple figures the time points are mixed/confused; etc.). Therefore, the paper should be reanalyzed and rewritten to make sure that the data is described properly.
Description of the scientific principles behind the experiments need to be more careful. For example, CD38 is not a costimulatory molecule (line 581); Superantigens bind Vb and MHC alpha chain of MHC, so without APC in the culture superantigens likely have no impact in the results (564-5).
English is ok.
Author Response
REBUTTAL:
Manuscript Number: Microorganisms - 2357988
Dear Reviewer
Thank you for the constructive comments. All the revised work on the manuscript has been highlighted in yellow.
Reviewer #3 :
Comments from the reviewer: The paper by Bhengu and colleagues maintained some major issues after the revision that still require the author’s attention, specifically in what regard the interpretation and description of the data and the scientific principles described. As an example, description of figure 1 is not clear at all, with graphs showing something and the authors describing something else (e.g., no significance in IFN-γ) between TB and Ascaris when the graph shows otherwise. lines 251-253 the authors contradict themselves; the use of tunnel and flow cytometry to confirm the data is not clear; in multiple figures the time points are mixed/confused; etc.). Therefore, the paper should be reanalysed and rewritten to make sure that the data is described properly.
The data was reanalysed, and results rewritten as seen below.
Figure 1. Data for IFN-γ, TNF-α, Granzyme B and perforin are shown for Jurkat cells at 24 and 48 hr time points. The unstimulated control cells and LPS-stimulated cells were negative and positive controls, respectively.
Figure 2. Data for IFN-γ, TNF-α, Granzyme B and perforin are shown for THP-1 cells at 24 and 48 hr time points. The unstimulated control cells and LPS-stimulated cells were negative and positive controls, respectively.
IFN-γ and TNF-α (at both 24 and 48 hour stimulation time points), Granzyme B (24 hour stimulation only) and perforin (48 hour stimulation only) levels were significantly higher in the TB singly stimulated Jurkat cells compared to the unstimulated controls, LPS, Ascaris and Ascaris plus TB costimulated Jurkat cells cells (p < 0.0001) (Figure 1). Similar results were noted for the THP-1 stimulated cells, apart for perforin where similar findings were noted at 24 hours and 48 hours (Figure 2).
Figure 3. Jurkat cell line responses for Th1/pro-inflammatory (IL-2 and IL-17) and transcription factors (Eomes and NFATC2) at 24 and 48 hr timepoints. The unstimulated control cells and LPS-stimulated cells were negative and positive controls, respectively.
IL-2, IL-17, Eomes and NFATC2 (at both 24 and 48 hour stimulation) were significantly higher in TB singly stimulated Jurkat cells compared to the unstimulated controls, LPS, Ascaris and TB plus Ascaris stimulated cells (p < 0.0001) (Figure 3).
Figure 4. THP-1 cell line responses for Th1/pro-inflammatory (IL-2 and IL-17) and transcription factors (Eomes and NFATC2) at 24 and 48 hr time points. The uninfected control cells and LPS-stimulated cells were negative and positive controls, respectively.
IL-2, IL-17, Eomes and NFATC2 (at both 24 and 48 hour stimulation) were significantly higher in TB singly stimulated Jurkat cells compared to the unstimulated controls, LPS, Ascaris and TB plus Ascaris stimulated THP-1 cells (p < 0.0001) (Figure 4).
Figure 5. IL-4 and IL-5 responses in TB, Ascaris, and dually stimulated Jurkat and THP-1 cells at 24 and 48 hr time points.
IL-5 (24 and 48 hour stimulation) levels were significantly lower in the TB singly stimulated cells compared to the Ascaris singly stimulated and TB plus Ascaris costimulated Jurkat and THP-1 cells, which had similar expression levels (p < 0.0001). Similar findings were noted for IL-4, however the Ascaris singly stimulated cells had significantly higher IL-4 levels compared to the TB plus Ascaris costimulated Jurkat (48 hour stimulation) and THP-1 cells (24 and 48 hour stimulation) (p < 0.0001.
Figure 6. Jurkat and THP1 cell responses for regulatory cytokines (TGF-β, IL-10 and FoxP3) at 24 and 48-hour time points. The uninfected control and LPS-stimulated cells were used as negative and positive controls, respectively.
TGF-β levels were significantly lower in the TB singly stimulated cells Jurkat cells (24 hour stimulation) compared to the TB plus Ascaris costimulated cells, however the opposite trend was observed for THP-1 cells (24 hour stimulation) (p <0.0001). Conversely, no significant changes were noted in Jurkat and THP-1 cells (48 hour stimulation) between the TB singly stimulated and TB plus Ascaris costimulated cells. IL-10 levels were significantly lower in the TB stimulated Jurkat (24 and 48 hour stimulation) and THP-1 (24 hour stimulation) cells compared to the TB plus Ascaris costimulated cells (p <0.0001). FoxP3 levels were also were significantly lower in the TB singly stimulated cells Jurkat and THP-1 cells (24 and 48 hour stimulation) in comparison to the TB plus Ascaris costimulated cells (p <0.0001). (Figure 6).
Figure 7. IFN-γ, TNF-α, IL2, IL-17, Perforin and Granzyme B ex vivo gene expression data.
IFN-γ, TNF-α and IL-17 were highest in TB infected group (albeit only six individuals). Perforin was similar across all groups, while Granzyme B levels differed between the control and the coinfected groups (p <0.0001), helminth and coinfected groups (p = 0.0020) and then between TB and coinfected groups (p<0.0001). IL-2 had different levels between the control and the TB plus helminth coinfection group (p<0.0001) and also between the helminth and coinfected group (p= 0.0067).
Figure 8. Eomes and NFATC2 gene expression values.
Eomes was higher in the controls than the TB/helminth co-infected (p <0.0001) and higher in TB compared to the coinfected individuals (p <0.0001). NFATC2 was significantly higher for controls compared to the coinfected individuals (p=0.0003) and higher between the helminth and the TB/helminth coinfected individuals( p=0.0032).
Figure 9. IL-4, IL-10, TGF-β and FoxP3 gene expression values.
IL-4, IL-10 and TGF-β were higher among uninfected controls and helminth infected individuals (albeit a wide distribution) and lower in the TB singly infected and coinfected groups (albeit a small sample size). TGF-β was low in the TB singly infected group compared to the controls (p=0.0012) and between the controls and the coinfected individuals (p<0.0001). FoxP3 was significantly lower among the TB infected compared to both the control (p<0.0001) and helminth-infected groups (p=0.0012).
Description of the scientific principles behind the experiments need to be more careful. For example, CD38 is not a costimulatory molecule (line 581)
CD 38 has been described as an immunomodulatory and not a costimulatory molecule.
Superantigens bind Vb and MHC alpha chain of MHC, so without APC in the culture superantigens likely have no impact in the results (564-5).
A superantigen has been identified in TB, which does not require antigen processing by antigen presenting cells. This was determined by the study which aimed to better understand the T cell populations that play a role in the immunopathogenesis of human tuberculosis, by studying the TCR chain repertoire displayed in tuberculous pleuritis patients. The possibility that Mtb has a superantigen was raised by the findings of the polymerase chain reaction and flow cytometry analysis, which revealed an expansion of Vβ8+ T lymphocytes near the site of illness in some patients. In vitro, Mtb strongly stimulated the proliferation of T cells in tuberculin-negative healthy participants, with preferential Vβ8+ T cell growth independent of the CDR3 region—the antigen-presenting cells were unnecessary for T cell stimulation, which was MHC class II-dependent.

Round 3
Reviewer 3 Report
The presentation of the data is now much clearer.
A few points need further clarification:
"The present study demonstrated a typical TB response characterized by an increase in inflammatory cytokines such as IFN-γ, TNF-α, IL-2 and IL-17. However, we did not use immunomodulatory molecules such as anti-CD 28 to stimulate the Jurkat cells, since they do not possess antigen presenting properties. Therefore, the Jurkat cell response may be suboptimal, due to the exclusion of immunomodulatory molecules and this is a limitation for this study."
1- CD28 is a costimulatory molecule (CD38, as it was described in the previous version of the manuscript is not a costimulatory molecule).
The main issues of this approach (and the description above) is the absence of APCs in this system. Therefore, the jurkat response is suboptimal due to the absence of costimulation and antigen presentation.
"A superantigen has been identified in TB, which does not require antigen processing by antigen presenting cells (...)"
2- The superantigen, while not requiring processing by APCs, still requires MHC-II, as the author state and reference in the text. In this regard, it is not clear how a superantigen can influence the response of jurnkat cells when there is no APCs (and therefore MHC-II in the culture). As the author's state (and reference) the effect of the putative supeantigen is MHC-II dependent.
Author Response
REBUTTAL:
Manuscript Number: Microorganisms - 2357988
Dear Reviewer
Thank you for the constructive comments. All the revised work on the manuscript has been highlighted in yellow.
Reviewer #3 :
Comments from the reviewer: The presentation of the data is now much clearer.
A few points need further clarification:
"The present study demonstrated a typical TB response characterized by an increase in inflammatory cytokines such as IFN-γ, TNF-α, IL-2, and IL-17. However, we did not use immunomodulatory molecules such as anti-CD 28 to stimulate the Jurkat cells, since they do not possess antigen-presenting properties. Therefore, the Jurkat cell response may be suboptimal, due to the exclusion of immunomodulatory molecules and this is a limitation for this study."
CD28 is a costimulatory molecule (CD38, as it was described in the previous version of the manuscript is not a costimulatory molecule).
CD 38 was a typing error and has been amended. CD28 is a costimulatory molecule that is explained in the discussion and the limitation section.
The main issue of this approach (and the description above) is the absence of APCs in this system. Therefore, the Jurkat response is suboptimal due to the absence of costimulation and antigen presentation.
The explanation has been provided in lines 496 – 501 (see below)
The present study demonstrated a typical TB response characterized by an increase in inflammatory cytokines such as IFN-γ, TNF-α, IL-2, and IL-17. However, we did not use costimulatory molecules such as anti-CD 28 or anti-Cd49d to enhance the stimulation of the Jurkat cells, since they do not possess antigen-presenting properties. Therefore, the Jurkat cell response may be suboptimal, due to the exclusion of immuno co-stimulatory molecules, which is a limitation of this study.
"A superantigen has been identified in TB, which does not require antigen processing by antigen-presenting cells (...)"
2- The superantigen, while not requiring processing by APCs, still requires MHC-II, as the author state and reference in the text. In this regard, it is not clear how a superantigen can influence the response of Jurkat cells when there are no APCs (and therefore MHC-II in the culture). As the authors state (and reference) the effect of the putative superantigen is MHC-II dependent.
The explanation about the superantigen has been removed and replaced with the statement below.
The present study demonstrated a typical TB response characterized by an increase in inflammatory cytokines such as IFN-γ, TNF-α, IL-2, and IL-17. However, we did not use costimulatory molecules such as anti-CD 28 or anti-Cd49d to enhance the stimulation of the Jurkat cells, since they do not possess antigen-presenting properties. Therefore, the Jurkat cell response may be suboptimal, due to the exclusion of immuno co-stimulatory molecules, and this is a limitation of this study.
